# Differences among Male and Female Spanish Teachers on Their Self-Perceived Preparation for Inclusive Education

**DOI:** 10.3390/ijerph19063647

**Published:** 2022-03-18

**Authors:** Natalia Triviño-Amigo, Sabina Barrios-Fernandez, Carlos Mañanas-Iglesias, Jorge Carlos-Vivas, José Carmelo Adsuar, Ángel Acevedo-Duque, Jorge Rojo-Ramos

**Affiliations:** 1Social Impact and Innovation in Health (InHEALTH) Research Group, Faculty of Sport Sciences, University of Extremadura, 10003 Cáceres, Spain; natalia.trivino@alu.uhu.es (N.T.-A.); cmaanasi@alumnos.unex.es (C.M.-I.); 2Promoting a Healthy Society Research Group (PheSO), Faculty of Sport Sciences, University of Extremadura, 10003 Cáceres, Spain; jorgecv@unex.es (J.C.-V.); jadssal@unex.es (J.C.A.); 3Public Policy Observatory, Universidad Autónoma de Chile, Santiago 7500912, Chile; angel.acevedo@uautonoma.cl

**Keywords:** inclusive education, special needs, disability, teacher preparation, gender, perceptions

## Abstract

Inclusive education is a right for every student, being one of the current challenges with which the education system is struggling. The teacher’s role in this process is essential in building an inclusive and transformative school. This study aims to measure Spanish secondary school teachers’ perceptions of their preparation to address inclusive education, exploring whether there are differences concerning their gender. A total of 420 Spanish secondary school teachers responded to three dichotomous questions about their initial and ongoing preparation using the Evaluation of Teacher Education for Inclusion Questionnaire (CEFI-R). The results suggest that there are no significant differences between men and women regarding their perceived readiness to deal with diversity. However, women seem to be more confident in their competence, as they show less need for preparation in addressing the diversity of needs in their students with disabilities, and in promoting inclusive education.

## 1. Introduction

Education systems must adapt according to prevailing social paradigms and changes; in the current social context where rights and quality of life are at the forefront, schools must transform themselves by embracing inclusive values [1,2]. Inclusive education is defined as the response to diversity in positive ways, characterized by participation, achievement, and the elimination of barriers for all groups involved [3,4]. Among the various challenges are the following: the provision of adequate and personalized attention to students with special educational needs, the establishment of reasonable supports necessary to maximize their learning, and social participation in the educational setting [5]. Inclusive education means that all children have the right to receive a quality education, as schools must not only teach knowledge but also educate the citizens of the future, therefore, educational stakeholders must coordinate and play the necessary roles to achieve real transformative schools [4,6,7].

The establishment of an inclusive education system is an ambitious but necessary goal, constituting one of the Sustainable Development Goals (SDG) established for the 2030 Agenda by the United Nations (UN) [2,8]. This education statement has been promoted for more than forty years, through texts such as the Warnock Report [9], the Salamanca Statement [10], the Dakar Framework for Action [11], and the Index for Inclusion [4] highlighting the important role that schools play in achieving quality education for all, including students with disabilities. In Spain, the Organic Law 3/2020 which amends Organic Law 2/2006 on Education (LOMLOE) [12] is the current state educational legislation. The Spanish education system is divided into several levels: the preschool stage for children up to 6 years (non-compulsory); primary education (compulsory), six academic courses for children between 6 and 12; secondary education comprises compulsory secondary education from 12 to 16 (last compulsory stage), followed by professional education or baccalaureate. In order to regulate a focus on diversity, each region must enact its decrees, adapted to its distinctive characteristics. In Extremadura, the Spanish region where the study took part, the 228/2014 Decree on Attention to Diversity [13] regulates educational supports for students with specific and special needs. In 2018, there were 219,720 students with special educational needs (32.9%), representing 1.6% of the total student body in Extremadura [14].

Teachers are necessary agents in achieving inclusive schools, as they must organize the educational response by designing, and developing individualized educational practices, with different methodologies, resources, and technologies, encouraging the motivation, participation, and learning of all students, including those with disabilities [15,16]. In addition, together with all other educational stakeholders, they must overcome the barriers to an inclusive school, which include inflexible teaching systems focused on conceptual content, the absence of shared responsibility among educational agents, and the lack of leadership, which can lead to frustration and helplessness among teachers [17,18]. A teacher’s preparation is key to an inclusive and transformative school [16,19,20], and this includes their initial and ongoing development. There are two factors directly related to teachers’ success in addressing diversity, and promoting an inclusive school: (a) their preparation, both initial and ongoing [21], and direct contact with students with special educational needs [22,23]. Thus, to acquire these competencies, the initial preparation of future teachers should include practical experience with students with special needs and teachers with experience in inclusive settings [22]. This leads us to the concept of attitude, i.e., the predisposition towards a certain behavior concerning a group or phenomenon [5,24], and about which the literature warns that some teachers would not be receptive to diversity [25]. Another relevant factor is self-efficacy, a personal belief in one’s abilities in a given situation [26], which is considered a mediator of attitudes [27]. Thus, teachers with high levels of self-efficacy report higher levels of work satisfaction, while lower levels are related to work-related stress and difficulties in dealing with students’ misbehaviors [28]. For all of the above reasons, the university is called to profoundly reflect upon the curriculum and the attitudes on which it is currently operating, and to decide whether these correspond to the demands of educational inclusion, based on a paradigm of rights, citizenship, and quality of life [7,8].

In addition to the aforementioned, gender-sensitive research is currently required. Therefore, we aim to check whether gender is a factor that influences Spanish secondary school teachers’ perceptions of their preparation for inclusive education. Studies have found that gender does not seem to affect attitudes, although some studies have found better attitudes towards disability, in female educators [29]. The literature reports that more research is needed concerning self-efficacy [28]. Regarding teachers’ perceptions of their preparation, no studies are available to the best of our knowledge. Thus, our study aims to investigate the perceived preparedness of secondary school teachers regarding diversity and inclusion, and to investigate whether there are differences between men and women.

The following hypotheses are proposed:

**Hypothesis** **1** **(H1).**
*Significant sex differences will be found in the perceived initial and ongoing preparation for inclusive education and their willingness to attend specific courses on inclusive education.*


**Hypothesis** **2** **(H2).**
*Females will score higher than males on all dimensions of the CEFI-R questionnaire, meaning they have lower perceptions of preparation needs for addressing diversity and promoting inclusive education.*


## 2. Materials and Methods

### 2.1. Participants

The sample consisted of 420 Spanish secondary school teachers working in public schools in the region of Extremadura. A non-probability sampling method based on convenience was used [30].

Participants were asked about several sociodemographic data: sex, age, province of the school in which they work, and years of teaching experience. This information can be found in Table 1. In addition, comment on the median years of teachers’ experience was 15 (interquartile range = 19).

### 2.2. Procedure

This trial was conducted between September 2020 and July 2021. To access the sample, an e-mail was sent to all secondary education centres, which included the study aim, informed consent form, and a link to answer the questions using Google Forms (Google, Mountain View, CA, USA), composed of five sociodemographic questions, three dichotomous questions about their initial and ongoing preparation to face diversity and The Teacher Preparation for Inclusion Evaluation Questionnaire (CEFI-R) [31,32]. It was estimated that participation in the study will require only about 10 min, which we consider essential to get the widest possible sample, given the current situation of teacher overload and the challenges teachers are facing due to the pandemic.

All data were collected anonymously and kept private. The study was performed according to the guidelines of the Declaration of Helsinki and was approved by the Bioethics and Biosafety Committee of the University of Extremadura (protocol code: 184/2021).

### 2.3. Instruments

Participants were asked three dichotomous questions about their initial and ongoing training: (1) Do you think that you were properly prepared through your initial preparation to respond to the diversity of your students’ needs? (2) Has ongoing preparation helped you to respond to the diversity of your students’ needs? (3) Would you be willing to attend courses on inclusive education? These basic questions are intended for further comparison between different educational actors and educational stages.

The Evaluation of Teacher Training for Inclusion Questionnaire, CEFI-R was used [31,32]. The CEFI-R is made up of 19 items grouped into four dimensions: (1) Conception of diversity (5 items), which measures belief in the concept of diversity, place, and the form of schooling for students, and educational policy on diversity; (2) Methodology (5 items), for aspects relating to the design and development of an inclusive curriculum; (3) Supports (4 items), about a teacher’s perception and role in this concept; and (4) Community participation (5 items), which measures collaboration between all educational stakeholders. This instrument uses a Likert scale where the values range from 1 to 4, ranging from 1 “Strongly disagree” to 4 “Strongly agree”. The authors reported a Cronbach’s alpha value of 0.79, with each factor above the 0.70 threshold. In a subsequent analysis, a Confirmatory Factor Analysis (CFA) of this model with four dimensions (Conception of Diversity, Methodology, Supports and Community Participation) was kept, with good and excellent goodness-of-fit values, and reliability measured with Cronbach’s Alpha was between 0.75–0.94, considered excellent [33].

### 2.4. Statistical Analysis

The Statistical Package for Social Sciences (Version 23, IBM SPSS, Chicago, IL, USA) was used. The Kolmogorov-Smirnov test was carried out to determine if the data followed a normal distribution. As this assumption was not met, for both the dichotomous questions (Kolmogorov-Smirnov = 0.484; *p* = 0.000; and Bartlett’s = 36.607; *p* = 0.000) and the CEFI-R items (Kolmogorov-Smirnov = 0.300; *p* = 0.000; and Bartlett’s = 3352.626; *p* = 0.000), non-parametric tests were chosen. Pearson’s Chi-square test was performed to analyze the differences between the three dichotomous questions according to teachers’ sex and educational stage. The Mann-Whitney U test was used to analyze the differences between the responses to the dichotomous questions and the CEFI-R dimensions. This test was also used to analyze the differences between the dimensions according to the sex of the teachers and the educational stage. The Spearman Rho was performed to explore the association between the CEFI-R dimensions and age. Cronbach’s Alpha was used to calculate the reliability of each of the dimensions from the CEFI-R. Reliability values between 0.60 and 0.70 were considered acceptable, while values between 0.70 and 0.90 are excellent [33].

## 3. Results

Table 2 shows the frequency distribution of the responses to the three dichotomous questions about teachers’ perceptions of their initial and ongoing preparation for inclusive education, according to sex. Pearson’s Chi-Square test was used to analyze the differences.

Table 3 shows the distribution of frequencies and differences in each dimension of the CEFI-R according to the responses to the three dichotomous questions. The Mann-Whitney U test was used to analyze the differences. In Dimension (1) Conception of Diversity, statistically, significant differences were found according to the answers to the three questions. In Dimensions (2) Methodology, and (3) Supports, statistically significant differences were found according to the answers to questions 2 and 3. Finally, in Dimension (4) Community Participation, statistically significant differences were found according to the answers to questions 1 and 3.

The CEFI-R Dimension scores (Conception of Diversity, Methodology, Supports, and Community Participation) are shown in Table 4. Although males scored slightly higher than females in Dimensions (1) and (2), the difference was not statistically significant. In Dimension (3), although females scored slightly higher than males, the difference was not significant. In Dimension (4) the scores were identical for both sexes and were not significant either.

The correlation coefficient between the Dimensions and the different age ranges was carried out using the Spearman test (Table 5). The results show that Dimension (1) Conception of diversity, and (4) Community Participation, were significant.

## 4. Discussion

### 4.1. Main Findings and Theoretical Implications

This research describes secondary school teachers’ perceptions of their preparation for inclusive education in the Spanish context and their preparation needs in this field. For this purpose, the CEFI-R questionnaire was used, a validated multidimensional instrument composed of 19 items grouped into four dimensions (Conception of diversity; Methodology; Support; Community participation). Furthermore, three dichotomous questions about their initial and ongoing preparation were asked, finding no significant differences between the sexes.

Regarding the acquisition of necessary competencies in their initial preparation, 79.3% (*n* = 333) of teachers answered that they did not receive adequate preparation to respond to diversity. Along these lines, Vélez-Calvo [34] concluded that, despite the one-year increase in the length of education degrees in Spain, there was no increase in teachers’ inclusive education preparation. According to our findings, 79.2% (*n* = 299) of teachers think that ongoing preparation helped them to address diversity and 81.9% (*n* = 344) would be willing to receive specific training on inclusive education. This is in line with some studies reporting the importance of professional development for teachers [35,36,37]. A previous study found that 31.7% of teachers perceived that they were prepared in their initial preparation [38]. Questions 2 and 3 showed that 86.6% of teachers perceived that ongoing preparation helped them, and 95.8% would be willing to receive specific training on inclusive education, respectively. In the study requirements of prospective teachers, it was found that 55.5% felt their initial preparation enabled them to properly address diversity [39].

Based on the CEFI-R questionnaire scores, the following was found. In Dimension (1) Conception of diversity, a median of 3 out of a possible 4 points was obtained, finding significant differences between teachers who would attend ongoing preparation and those who would not. No significant differences were found between the sexes. However, considering Mondragón-Barrera’s specified values [40], a mean inverse correlation (r = −14) was found between Dimension 1 and the age variable. The results in this dimension are in line with other studies on the Conception of Diversity [41,42,43], suggesting that teachers showed a willingness to work inclusively in classrooms [44]. In Dimension (2) Methodology, the median score was 3 points, with no significant differences between the sexes. However, teachers who did not feel that their ongoing preparation had helped them to address diversity, scored significantly lower. This contrasts with studies that show that adequate planning, curriculum revision, and flexible use of methodologies are essential to address the needs of students with disabilities [23,45]. The results for Dimension (3) Support, were the lowest (Me = 2.2) in the study, in line with the results obtained in another study performed with Spanish primary school teachers using the CEFI-R (Me = 2.4) [38]. However, other studies got better scores in this dimension with future teachers (Me = 3) [46], in accordance with the findings provided by Perlado [47]. This work highlights the importance of the collaboration between specialist teachers and classroom teachers to promote the successful development of students with special educational needs. Again, no significant differences were found in terms of sex or age. Finally, Dimension (4) Community participation got the highest results in the study (Me = 3.6), in line with [36,40] and showing significant differences between this dimension and teachers who would be willing to attend ongoing preparation. As Arnáiz [17] found, schools where the community was more involved and working together, showed a better predisposition towards inclusive education and better learning outcomes in all areas. A mean inverse correlation (r = −0.12) was found between Dimension 4 and the variable age [40]. Findings regarding the inverse correlation between the variable age and the Dimensions (1) Conception of diversity and (4) Community participation could have implications for older teachers because of the importance of perceived self-efficacy in coping with the challenges included in their new roles and in collaborating with other educational agents [48].

### 4.2. Practical Implications

Among the practical implications of this work, the following can be listed: (1) to support the development of educational regulation and legislation that helps educational agents make decisions on the provision of supports, methodological adaptations, and resources to achieve an inclusive school, following the United Nations Declarations [2,7] and the Sustainable Development Goal 4: “Quality education” [8]; (2) to carry out a thorough reflection on the contents and competencies in the initial education of teachers, weighing up whether they are in line with the current needs of the school; (3) to offer teachers ongoing preparation to address diverse educational inclusion dimensions; (4) to consider the establishment of a system for monitoring teachers’ ongoing preparation, given that until now, this has been voluntary; (5) to promote channels of collaboration between the school and the community so that all the agents involved can work together to achieve a transformative school and society.

### 4.3. Limitations and Future Lines

This study presents several limitations. The results should be interpreted with caution as the sample size was limited, and it was drawn from the same region of Spain, with the possibility of socio-cultural and/or legislative differences in other Spanish regions, as education is delegated to Autonomous Communities. Hence, it was a convenient sample that was used (non-probability sampling). No questions were included about the specific courses that participants took during their initial or ongoing education, so it is not possible to predict which type of courses enhance readiness for inclusion. In this study, online questionnaires were used for data collection, as they offer advantages such as reducing costs and facilitating data collection and processing [49]. Challenges focus on sampling, response rate, and non-respondent characteristics [50]. In the future we intend to extend this study to other regions, as well as to involve other educational stages such as early childhood education or post-compulsory education.

## 5. Conclusions

On the basis of our findings, both men and women consider that their initial preparation for inclusive education is not enough. They are aware that they need additional preparation, and they are willing to attend further ongoing preparation courses on diversity and inclusive education. No statistically significant sex differences were found. However, women seem to be more confident in their competence, as they express less need for preparation to address the diversity of their students with disabilities and to promote inclusive education.

Thus, concerning the hypotheses raised at the beginning of the study:

**Hypothesis** **1** **(H1).**
*Significant sex differences will be found in the perceived initial and ongoing preparation for inclusive education and their willingness to attend specific courses on inclusive education.*


This hypothesis is rejected.

**Hypothesis** **2** **(H2).**
*Females will score higher than males on all dimensions of the CEFI-R questionnaire, meaning they have lower perceptions of preparation needs for addressing diversity and promoting inclusive education.*


This hypothesis is accepted.

## Figures and Tables

**Table 1 ijerph-19-03647-t001:** Sample sociodemographic data (*n* = 420).

Variable	Categories	*n*	%
Sex	Men	153	36.4
	Women	267	63.6
Age (years)	Under 30	42	10
	31–40	96	22.9
	41–50	145	34.5
	Over 51	137	32.6
Position	Internym	137	32.6
	Official	283	67.4
School province	Cáceres	188	44.8
	Badajoz	232	55.2

*n* = number; % = percentage.

**Table 2 ijerph-19-03647-t002:** Distribution of the three dichotomous questions according to sex.

			Yes	No	*p*
(1) Do you think that you were properly prepared through your initial preparation to respond to the diversity of your students’ needs?					
Sex	Men	*n* (%)	32 (20.9)	121 (79.1)	0.93
	Women	*n* (%)	55 (20.6)	212 (79.4)	
	Total	*n* (%)	87 (20.7)	333 (79.3)	
(2) Has ongoing preparation helped you to respond to the diversity of your students’ needs?					
Sex	Men	*n* (%)	108 (70.6)	45 (29.4)	0.83
	Women	*n* (%)	191 (71.5)	76 (28.5)	
	Total	*n* (%)	299 (71.2)	121 (28.8)	
(3) Would you be willing to attend courses on inclusive education?					
Sex	Men	*n* (%)	121 (79.1)	32 (20.9)	0.25
	Women	*n* (%)	223 (83.5)	44 (16.5)	
	Total	*n* (%)	344 (81.9)	76 (18.1)	

*p* of the Pearson’s Chi-square test.

**Table 3 ijerph-19-03647-t003:** Analysis of the CEFI-R questionnaire differences for every dimension according to the answers to the dichotomous questions.

	Question 1			Question 2			Question 3		
Dimensions	Yes	No	*p*	Yes	No	*p*	Yes	No	*p*
1. Conception of Diversity	3 (1.2)	3 (1.2)	0.10	3 (1.2)	2.8 (1.2)	0.03	3 (1)	2.4 (1)	<0.01
2. Methodology	3 (1.4)	3 (1.1)	0.26	3 (1)	2.8 (1)	0.01	3 (1.2)	3 (1.1)	0.42
3. Supports	2.2 (1)	2.2 (0.8)	0.27	2.2 (0.8)	2.2 (0.9)	0.47	2.4 (0.8)	2 (0.8)	<0.01
4. Community Participation	3.6 (1)	3.6 (0.8)	0.65	3.6 (0.8)	3.6 (1)	0.98	3.6 (0.8)	3.3 (1.4)	<0.01

Me = Median Value; IQR = Interquartile Range. Each score obtained is based on a Likert scale (1–4): 1 being “Strongly Disagree”, 2 “Partially Disagree”, 3 “Partially Agree”, and 4 “Strongly Agree”.

**Table 4 ijerph-19-03647-t004:** CEFI-R descriptive analysis and differences of each dimension, searching for differences between sex and educational stage.

	Total	Sex		
Dimensions	Me (IQR)	Men	Women	*p*
1. Conception of Diversity	3 (0.4)	3 (1.2)	3.4 (1)	0.13
2. Methodology	3 (1.2)	3 (0.9)	3 (1.4)	0.66
3. Supports	2.2 (0.8)	2.2 (1)	2.4 (1)	0.31
4. Community Participation	3.6 (0.8)	3.6 (1)	3.8 (1)	0.87

Me = Median Value; IQR = Interquartile Range. Each score obtained is based on a Likert scale (1–4): 1 being “Strongly Disagree”, 2 “Partially Disagree”, 3 “Partially Agree”, and 4 “Strongly Agree”.

**Table 5 ijerph-19-03647-t005:** Correlations between the Dimensions and the age group variable.

Dimensions	Age *ρ (p)*
(1) Conception of diversity	−0.14 (<0.01 **)
(2) Methodology	0.08 (0.08)
(3) Supports	−0.06 (0.21)
(4) Community Participation	−0.12 (<0.01 **)

The correlation is significant at the ** *p* < 0.01. Each score obtained on the Dimensions is based on a Likert scale (1–4): 1 being “Strongly Disagree”, 2 “Partially Disagree”, 3 “Partially Agree”, and 4 “Strongly Agree”.

## Data Availability

The datasets used during the current study are available from the corresponding author on reasonable request.

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
