# Peer review of "Differences among Male and Female Spanish Teachers on Their Self-Perceived Preparation for Inclusive Education"

_ijerph, 2022, doi:10.3390/ijerph19063647_

Round 1
Reviewer 1 Report
GENERAL COMMENTS:
The manuscript was written according to the journal´s template and with appropriate sections.
In my opinion, the paper is too short for full article, it lacks information and is presented in an approximate way in the Materials and Methods part. Furthermore, the novelty of the work presented is not evident. My observation is that more in-depth future research is needed.
Explanation
This paper needs some important revisions. The authors to better describe each part of the work presented in such a way that it is clear to the reader in order to facilitate its revision:
- Materials and methods – 2.3. Instruments
The description of this part is confusing. More details are needed about three questions and four dimensions, used in the study. Here, details are missing and the authors should better describe.
- Results – Table 1: I think the Age need to be corrected as
Under 30
Between 31 and 40
Between 41 and 50
Over 51
- Conclusion - This section authors could be improved. In conclusion, point out which are the scientific and which are the social justifications of the research. Emphasize the importance of this approach.

Author Response
Dear Reviewer,
Thank you for your review of our manuscript. We have carefully considered your comments and believe that the quality of the paper has improved after incorporating your suggestions.

Reviewer 2 Report
My comments are found in the attached Word file below

Author Response

(The authors gave the same response as above.)

Reviewer 3 Report
There were no hypotheses? There is a need to develop clear hypotheses and having justification from state of the art to justify knowledge gap for the study. Which theoretical lens is relevant to the research study? The manuscript should document the main contribution of the study. Currently the question responses are individually discussed, the dimensions should be used to develop a model which can be mathematically evaluated. The model should be grounded in the theoretical lens.
Author Response

(The authors gave the same response as above.)

Round 2
Reviewer 1 Report
After reviewing the paper corrections, I have the following recommendation:
Paper acceptance, without any other corrections.
Author Response
Dear Reviewer (1),
Thank you for your review of our manuscript.
Reviewer 2 Report
The Discussion section looked more like a presentation of results than a real discussion of results. The authors should show how the results interract with previous and current relevant literature or research. None of that is shown in the discussion section.
Author Response
Dear Reviewer (2),
Thank you for your review of our manuscript. We have carefully considered your comments and
believe that the quality of the paper has improved after incorporating your suggestions. Below are our
responses to your suggestions:
Authors’ reply: Thank you very much. We have expanded the discussion to
respond to your comment.
Reviewer 3 Report
I would like to thank authors for the improvement in paper, however I still think following things can be improved.
1, Still the paper does not highlight the theoretical lens, it is important to orient the study with a scientific theory to understand the findings.
There is need to add three sections with practical implications, theoretical implications and limitations of the study.
Author Response
Dear Reviewer (3),
Thank you for your review of our manuscript. We have carefully considered your comments and
believe that the quality of the paper has improved after incorporating your suggestions. Below are our
responses to your suggestions:
Authors’ reply: Thank you very much. We have expanded and reorganised the
discussion to respond to your comment.